# Towards Decolonising Research Ethics: From One-Off Review Boards to Decentralised North–South Partnerships in an International Development Programme

**Maria Josep Cascant Sempere** [1,*] **, Talatu Aliyu** [2] **and Cathy Bollaert** [2]

1    Sociology & Social Anthropology Department, Open University, Milton Keys MK7 6AA, UK
2    Christian Aid, London SE1 7RT, UK; taliyu@christian-aid.org (T.A.); cbollaert@christian-aid.org (C.B.)
*    Correspondence: kas.sempere@open.ac.uk

**Abstract:** Contemporary North–South research collaborations are fraught with power relations originating in colonialism. Debates about research ethics have tended to turn around the "procedural ethics" formal model and the "everyday ethics" practical model. We build on that to suggest a second debate that scrutinises ethics and power relations not only in the researcher–researched relationship but also in the relationships within research teams and ethics review boards. The research asked: how can we shift power in research to decolonise research and build more equitable partnerships? We explored this with data obtained through collaborative autoethnography in a multi-country development research programme, Evidence and Collaboration for Inclusive Development (ECID). This included regular self-reflective meetings, visual methods, a self-evaluation survey, and blogs addressing power issues. Coordinated from London, the research had all the cards to adopt a 'colonial' gaze in which the North would 'research' the South. The case narrates the journey of the research team to decentralise power in the programme, which included sharing control over the selection of research topics, and the research design, budget, and publications. Drawing from the lessons learned from the research approach that was adopted in ECID, this paper offers an 8-step model towards decolonising research ethics.

**Keywords:** ethics; research; international development; decoloniality; knowledge production; ethics review boards; Myanmar; Nigeria; Zimbabwe; UK





## 1. Introduction

Many research consortiums and partnerships in the international development sector are financed by donors based in the global North (The "Global North" refers to countries with a high Human Development Index. They are industrially, technically, and socially well-developed. Most, but not all, are in the northern hemisphere. The "Global South" refers to countries with medium and low human development index, mostly based in South and Central America, Africa, and Asia.) The difficulty with this is that it can lead to research partnerships that are driven and decided by countries in the global North—from the research design and implementation phase to the interpretation and communication of the findings. These practices have received critiques of neo-colonialism [1] or coloniality [2], which can be understood as the long-term patterns of power and inequality that emerged because of colonialism and that continue to persist beyond the strict limits of colonial administrations. Such critiques argue that such practices lead to inequitable research partnerships and inequalities relating to whose questions are being asked, who produces and uses the knowledge, and what cultural and material inequalities are created and sustained during the research process. Critiques about knowledge inequalities are far from new yet they continue to receive strong public attention. For instance, in South African universities there have been numerous students' protests calling for justice in how knowledge is gener-

ated and for the elimination, or at least mitigation, of the disproportionate legacy of white European thought and culture [3].

Working as a researcher within the development sector often requires working within multi-country research and funding structures which do not promote fair and equitable research partnerships. Donor countries, which are usually based in the global North, tend to control the research agenda, with countries based in global South countries quickly become the sites for data collection with little say in the research agenda. However, within such partnerships there are three inequalities that stand up as particularly important, namely inequalities relating to who conceptualises and designs the research and defines the research roles; who controls the research budget; and who publishes the research. From a decolonial perspective, and in the interests of justice in knowledge production, all these aspects of research should be seen as an integral part of doing research ethically. In its broad conception, an ethical research process should create egalitarian research partnerships which promote and nurture the research ecosystem (universities, research councils, research careers, budget distributions, publication structures) of all the countries involved in the research. However, the problem is that ethics is not usually conceptualised or understood in this way and therefore a decolonial approach to ethics is not considered within a given research process. Instead, research ethics tends to take a "narrow" approach and focus on researcher–researched issues such as informed consent, the right to non-participation, confidentiality, and once-off approval processes by ethical review boards. Box 1 is an example of a typical ethical disclaimer that can be referred to as a "narrow" approach to ethics.

**Box 1.** Ethics disclaimer in a conventional approach to ethics.

> "This research project is being conducted by the University of X. Your participation in this study is voluntary. You may choose not to participate. If you decide to participate you may withdraw at any time. Your withdrawal will not be penalized. If you do decide to participate, your responses will be confidential and we will not collect identifying information except from your email address. The information you provide in this questionnaire will only be used by the research team. Your responses will be used in a way that ensures your anonymity. We will do our best to keep your information confidential. All data are stored in a password-protected electronic format. The results of this study will be used for scholarly purposes only. If you have any questions about the study, please contact X."

Source: own elaboration.

Although these processes are important and necessary, there are two reasons why this is considered a "narrow" approach to ethics: (i) from a technical perspective, it is focused on the legal aspects of research and not on its daily and contextual challenges; and (ii) from a political perspective, it is focused on the development of single projects and does not address the structural aspects of inequality and decoloniality in research and knowledge production.

Firstly, technical limitations are outlined in the "procedural versus daily ethics" debate [4–6]. Critiques within this debate point out that conventional "narrow" approaches to ethics are often carried out too formally and do not consider the potential adverse effects of the research or any additional ethical dilemmas that may occur as the research unfolds. For example, in some countries, asking for written informed consent can be understood as a form of auditing and dangerous. Such an approach risks reducing research participation and the quality of the data that are collected [7,8]. Another example relates to the risks of proclaiming the marvels of informed consent, when, in reality, some participants may have little or no power to choose to participate and may be coerced into participating, or they will participate in the research due to their cultural norms and wanting to please the newcomers with their participation as hosts [9]. There are also many other daily ethical dilemmas that are often overlooked, such as accepting certain unequal gender norms while doing research; paying, or not paying, research volunteers for their time; or intervening on behalf of poorer research participants by, for example, funding a health treatment for a

participant. Overall, most research processes end up being much more complicated than what is usually encompassed in a standard ethics disclaimer.

Secondly, political limitations to conventional "narrow" approaches to ethics refer to the lack of sensitivity towards (de)coloniality, the global inequalities that continue to shape the international development sector, and the historic call for epistemic, cognitive, or knowledge justice [10–12]. Limiting ethics to aspects such as informed consent and anonymity cannot address these larger power relationships that go beyond a single project. Consequently, a "broader" approach to research ethics aims to reduce and/or eliminate knowledge inequalities not only with the researched but also between researchers and between researching countries in all the stages of the research cycle—from the research design phase to publishing and disseminating research findings.

This paper argues that for multi-country, North–South research projects to be truly ethical, they should address these two debates relating to the technical and political limitations of the research ethics. This applies to any type of scientific discipline, from science, technology, engineering, and mathematics (STEM) to the social sciences. This requires paying attention to the legal and practical aspects of the research, to the academic and social outcomes, and to the short-term, project-based ethical aspects and to those that aim to change unequal structures in the longer term. Moreover, the paper asks: how can we shift power in research and build more equitable partnerships?

Based on data obtained through collaborative autoethnography, this paper narrates the journey of the research team to decentralise power and address structural inequalities in a multi-country research project; in doing so, it presents a case study of a multi-country research project that was part of the Evidence and Collaboration for Inclusive Development (ECID) international development programme, which aimed to follow a broader decolonised approach to doing ethics in research. In this instance, "decolonising" is understood as the eradication of long-term patterns of power and inequality that originated in colonialism that continue to persist beyond the strict limits of colonial administrations [2]. The case study contributes to the growing body of literature on humanising and decolonising ethics and research. Illustrating how theory can apply to practice, it shows how to work towards knowledge justice even when research is funded by Northern donors—in this case, the UK Government.

The paper begins by introducing the case study, aspects of the data collection and analysis, and ourselves as writers. We then review what a decolonised approach meant for us in the different stages of the research. We conclude this article with a reflection on the challenges and implications surrounding a decolonised approaches to ethics in international research.

## 2. A Collaborative Autoethnographic Study of the ECID Research

ECID [13] was a four-year development programme working on citizen action for accountable governance in three basic services—education, health, and social services. It was funded by the UK Government through UK Aid Connect and delivered through a consortium of nine partners led from London by the international NGO Christian Aid. The programme was implemented by in-country partner organisations in Myanmar, Nigeria, and Zimbabwe. ECID included a research workstream to support the ECID objectives which aimed to improve basic public services and strengthen marginalised voices, civil society, and state accountability. The research aimed to contribute to academic knowledge, improve programme practice, and carry out evidence-based advocacy.

The methodology used to inform this paper draws on collaborative auto-ethnography— a multivocal approach in which several researchers create and interpret autoethnographic data [14]. This addresses the ethical issue of representing or appropriating the voice of others and therefore extends the scope of individual autoethnography [15,16].

The first stage of the methodology involved a research forum which comprised the researchers from the four countries. The forum included regular self-reflective group moments where how to shift power at each stage of the research was discussed. Visual

mapping exercises were also used, where, for example, we drew country stakeholder maps to plan a decentralized communication strategy for each country. This stage also included a forum self-evaluation survey on the extent to which we felt we had been able to shift power in the research process through the decentralized approach that was taken. This process was also accompanied by "critical friends" who were able to ensure that research diversity was maintained through the types of academic references that were used and by reviewing the relevance of the results for both international and national audiences.

The second stage of the collaborative autoethnography involved the three members of the broader global research team (the authors) writing this article. Author 3 (Dr Bollaert) and, later, Author 1 (Dr Cascant-Sempere) coordinated the global ECID research project. Author 2 (Ms. Aliyu) coordinated the Nigeria research strand. In keeping with a decolonised approach to research ethics, with the author composition we tried to reflect the principles raised in this paper. Firstly, we are a global and hybrid team. [Personal information is given here about the authors-omitted for peer review]. Although we are writing in a research system that biases Northern academia, we hope that the perspectives put forward in this paper contribute to shifting the research landscape to be more equitable in its promotion of knowledge justice.

During the research process, there were several meetings among the three authors to discuss how we were approaching issues of power and decoloniality. We wrote several blogs (Reflective blogs can be consulted on https://evidenceforinclusion.org/innovating-with-ethics-panel/, https://evidenceforinclusion.org/decentralising-research/, and https://evidenceforinclusion.org/decolonising-research/) and attended several international academic panels on the matter of ethics and decolonising ethics (We refer to the panel "Unsettling research ethics to promote progressive global social change" in the 2021 UK DSA conference, https://nomadit.co.uk/conference/85#10052), where we gathered recent publications and ideas that have helped shape our thinking. The authors used their personal experience of the ECID programme and research process to inform the analyses and approaches put forward in this paper.

Apart from unequal relationships related to coloniality, the authors recognise that other dimensions have shaped the inequality and power dynamics represented in this paper, including different educational levels (from doctors to non-academics), age, gender, and English fluency. While there are no quick solutions to such issues, we do note that English was used as the common language. Consequently, we had to ensure that meetings (all online) were facilitated using plain English and that discussions were supported with summarised notes in the text bar of the meeting software that was used.

## 3. Findings and Discussion: An 8-Step Model towards Decolonising Ethics in Research

In this section we review the different steps that were taken in response to the question underpinning the research: how can we shift power in research and build more equitable research partnerships? We argue that these steps form a model towards decolonising ethics in research. The steps include building a hybrid multi-country research team and ethics panel comprising both academics and practitioners, co-designing research, decentralising the research budget, and co-publishing reports and papers.

### 3.1. Building a Truly Global Partnership Structure

The first component of a model towards decolonising research ethics requires building a partnership structure that is representative of the countries involved in the research. When the ECID programme first began, the consortium only included UK research partners, namely the Research Team at Christian Aid UK (REL), the Open University, and a third research organisation that later left the consortium. Against this structure it was almost inevitable that research would be conducted in a way that would perpetuate knowledge inequalities, with UK institutions conducting research *on* what were meant to be partner countries—Myanmar, Nigeria, and Zimbabwe, and not *with* them. Within this structure it

would be considered standard practice for global North researchers to carry out research in (and on) the global South rather than researching with researchers from all the participating countries as equal partners.

To shift the structure towards a more equitable model required building a multi-country network. Through this approach, several Southern research partners joined the research team, including the Kachinland Research Centre in Myanmar, the Institute of Development Studies at the University of Nigeria, and the research organisation Poverty Reduction Forum Trust in Zimbabwe.

Ideally, building a global research team should start at the point of writing and submitting a bid. However, this remains a challenge as bids often have a short lead in time which is not conducive to a co-designed approach to writing bid proposals. In addition, it is not unusual for funders based in the global North to require renowned entities to lead and implement the research. This means that from the get-go research is designed within an unequitable research structure that reinforces western hegemonic power and does not lend itself to equitable partnerships. To overcome this impediment requires considering how research budgets can be decentralised.

### 3.2. Decentralising Budgets towards the Global South

The current research funding environment within the UK means that research budgets are often centralised in organisational headquarters (which are mostly based in London). In the interests of shifting power in research, this puts an onus on the headquarter organisation to decentralise its budgets towards the global South—the second component of a decolonised approach to research ethics.

In the ECID programme the original budget was mostly assigned to UK partners. Redirecting some of the financial resources to the research partners based in Myanmar, Nigeria, and Zimbabwe required negotiations with the funders. However, it was evident that securing the research partnerships prior to the negotiations contributed to a successful outcome. Nonetheless, this process was cumbersome and faced legal challenges due to the nature of the current funding ecosystem and the way financial accountability is structured. Although it was not possible to decentralise all the funds, the funding structure was altered to allow country research partners to be more autonomous. It was also important that at each stage in the process transparency about the budget status be maintained. Indeed, a session on research budgets was included in the research forum (see Section 3.5) that was convened.

### 3.3. A Multi-Country and Hybrid Research Team

Forming a multi-country and hybrid research team is the third component necessary for moving towards a more decolonised approach to research. When forming the research team, the ECID programme decided to include not only academics but also development practitioners working in the field of study, citizen action, and accountable governance in basic public services. Practitioners were drawn from Christian Aid's staff in Myanmar, Nigeria, and Zimbabwe. A hybrid team comprising both academics and practitioners was a necessary, although insufficient, condition to achieve research for impact that was grounded in practice and responsive to social needs.

The role of the academic research partners was to ensure quality and coherence in the research. Practitioners complemented the team by ensuring that the research was aligned with ECID's impact objectives. The practitioners also supported the academic partners with logistics and building relationships with key decisionmakers, such as local governments, to ensure the research results were used for practice, policy, and advocacy.

As a minimum requirement, it was necessary to have two core team members from each country in the research team—one from the academic partner and one from the practitioner partner. This was a particularly useful approach for building a good team dynamic, creating trust, and forming a common identity, which proved pivotal for providing and receiving honest peer feedback.

### 3.4. A Multi-Country and Hybrid Ethics Panel

In parallel with the formation of a hybrid research team, the fourth component of a decolonised approach to research is having an equally diversified ethics panel.

Given that the research team was a hybrid team comprised of NGOs, research institutes, and universities, not everybody was covered by academic ethics committees, as would be standard and required practice in universities. The concern was that without an ethics review board of sorts, the quality, accountability, and rigour of evidence would be jeopardised. It could also put the research at risk of not being published, given that most academic publications require research to have gone through an ethical assessment. To overcome this there were several options available:

One option was to go through the ethical review boards that were available to our academic partners (i.e., the Open University or the University of Nigeria). However, the difficulty with this option was that these boards only offered a conventional, administrative 'one-time' approach to ethics which a decolonial approach to ethics seeks to challenge. As such, this "narrow" approach to ethics did not represent the multi-country and professional-academic diversity required for building more equitable research partnerships.

A second option was to create a bespoke ethics panel through which power could be shifted and a more equitable approach to ethics could be adopted. This was the approach that the ECID programme ultimately selected. To achieve this, the selection of panel members was decentralised and decided by the research teams in each of the four represented countries. The panel included both academics and development practitioners. This was a particularly innovative approach as many research ethics panels, especially in universities, are only formed of academics. While academics provide expertise on research quality, they do not necessarily have the skills and experience that practitioners have to align and apply research in a development context. Similarly, practitioners do not necessarily have the research skills that academics bring. In this way, each provides a unique set of skills that together can contribute towards stronger and more effective development outcomes.

In the context of the ECID programme, the ethics panel was made up of six members. This included members with senior research profiles, including two professors (one from Nigeria and one from the UK), an academic from the UK with deep knowledge of Myanmar, and a director from a research institute in Zimbabwe. In addition, it included members with senior international NGO profiles (a Head of Programme Innovation and a Director of Policy and Communications). The ethics panel also included gender representation.

The purpose of the ethics panel was to provide support and act as a 'critical friend' to the research partners. Moreover, the panel contributed to three strategic moments in the process, namely: (i) when the research teams publicly represented their research proposals prior to carrying out the fieldwork; (ii) by providing written feedback on the literature reviews and the data collection tools that were being used by each team (including focus group discussions, interviews, and surveys); and (iii) through written feedback on the final drafts of the research reports and articles.

Their mandate was to provide ethical and research leadership as well as cultural and impact alignment. This process was both supported and informed by the ECID ethics framework, which provided guidance on issues relating to risk assessments, responsible data management, informed consent, safeguarding and protection, gender equality and social inclusion conflict sensitivity, vulnerable populations, and research ethics during COVID-19 [17]. (This framework was originally developed by consortium members Dr Jude Fransman, Dr Cathy Bollaert and Dr Emma Haegeman.) The ethics panel drew on this framework to assess the work of the research teams, which added to their own experiences of conducting research in the global South and of the role of research ethics committees.

### 3.5. Co-Designing Research–The Online Research Forum

A fifth component towards decolonising research in ECID was to decentralise not only the budget but also the decision-making. This was supported by an intensive six-week

online "research forum" to co-design the research and enable country researchers to decide on their own research expenses and processes.

The "forum" followed a training-planning approach. This differed from other courses that teach how to do research to students or professionals unfamiliar with research. Given that the research team already had advanced research skills, the goal was to help them design a large research plan collaboratively. This took place within a six-week window.

To facilitate this process of co-designing the research, an online course developed by Christian Aid and the Open University entitled Evidence for Development Professionals was used [18]. As the course is designed around the research process, it enabled the research team to collectively discuss and design the different components of their research in a step-by-step manner, from identifying the research objectives and research questions to learning how to write for publication, collect and analyse data, and communicate the research for impact. This processes also facilitated engagement on research ethics, budget, and timeline.

Each country team—formed by at least one researcher and one practitioner—decided on their specific research questions, methods, and timeline. The forum served to standardise certain aspects of the research so that it represented a truly multi-country research endeavour rather than separate country research pieces working in parallel. For instance, it was important to have an umbrella multi-country research question but also national research questions relevant for each country context—aiming for a balance between multi-country coherence and country autonomy.

The forum also served to ensure that all country representatives could review each other's work, rather than this happening bilaterally (and hierarchically) between the London-based coordination team and the global South countries, which would only serve to reinforce inequalities. Carrying out the process in this way resulted in three research proposals (one per country) prepared by the country research teams, and one proposal prepared by REL and the OU, which provided a country comparative study.

The research team self-evaluated the forum as part of the collaborative autoethnography, with 78% (seven out of nine team members) responding to the survey. From the results everybody (100%) said they would recommend the research forum format to co-design research and to integrate research into programmatic work. The forum was positively assessed as it helped "come out with an agile research proposal relevant to each country", "agree on common topics and focus for research", and "ensure that no issues are missed at each stage".

Other responses noted that "co-designing not only enhances ownership but more immersion into the whole program", that "it enables learning, sharing and coherence of ideas at the earliest stage of research conceptualization" and that "the comparative element across the three countries brings in an interesting and intriguing perspective". By contrast, the "language, context and translation at local level for Myanmar" was mentioned as a persisting challenge as well as the need for "more time between sessions to allow a thorough undertaking of 'homework' [this refers to the preparation that was required ahead of each session, e.g., designing research questions and drawing a stakeholder map]".

*3.6. Citing Global South Authors and Perspectives, and Oral Sources*

A sixth component to decolonise the research process relates to the question of whose knowledge counts and is included in research.

In the context of ECID, a recommendation put forward was to include a mixture of Northern and Southern perspectives in the literature reviews that framed the research [19]. Thus, the literature reviews sought to include and cite authors from the country or continent the research was taking place in. Arguably, this added to the quality of the research as diverse national and international perspectives on the topic of study were included thereby making the research more contextually and culturally appropriate—a central tenet to decolonising research.

Overall, the ECID research did not prioritise any methodology or methods. As Barnes [20] reflects on in "decolonising methodologies", engaging with power-aware epistemological standpoints (critical theory, decolonial theory, etc.) is more critical than using certain methodologies in terms of decolonising research. Similarly, Walker and Boni [21] suggest that using participatory research methodologies is not always a sufficient condition to claim epistemic (knowledge) justice. Overall, methods should be selected based on how well they help answer research questions and as long as critical ethics informs the research.

Further to this, the ECID research explicitly welcomed oral sources, which some journals are starting to acknowledge (see, for example, https://www.ajol.info/index.php/ajsw/about/submissions). Using such sources recognises the role of oral culture and methods in research or "orature" (unwritten literature) such as proverbs, idioms, songs, and stories which are integral to the social fabric of many societies living in the global South. By recognising this, the Zimbabwe team was able to include oral sources in its research outputs [22].

### 3.7. Publishing in Global South Journals

A seventh component in a model towards decolonising research ethics requires publishing in journals that are coordinated from an African, Asian, or Latin American organisation or university. Alternatively, if the journal is based in the global North, at a minimum it should be edited and/or peer-reviewed by an ethnically diverse and aware team.

This approach to publishing is central to shifting power in knowledge production and consolidating academic and publishing structures across different continents. It was recommended as a criterion in the selection of journals within the ECID research, together with other criteria such as thematic specialism and journal quality. (Myanmar went through a coup d'état during the research period. As a result, the country team decided to finalize the literature review stage but to discontinue the main primary data research and related publications).

In aggregate terms, research continues to be more abundant in richer countries [23], which results in a higher probability of academics from the global North having their work cited. Academics from the global South tend to have more limited opportunities to publish their work and would tend to publish in journals from the global North [23]. Moreover, research done in the global South is mostly published by Northern researchers with little co-authorship acknowledged to global South authors [24].

Another challenge surrounding the publication of research in southern based journals may be due to a cultural bias in thinking that journals based in the global North journals (and the papers they publish) are of better quality. This is not necessarily the case as many African, Asian, and Latin American journals compete in well-renowned international academic ranking bodies such as Scopus *and the* Web of Science. In some cases, alternative platforms have been created (see, for example, https://www.journalquality.info/en/). This happens in a context where non-indexed formats (such as working papers and policy reports) and non-English publications outside the main databases are not considered in academic circles, which creates knowledge inequalities [25].

These challenges relate more to the legacy of colonial power structures on knowledge production and use. However, in ECID there were also some conflicts of interest when it came to publishing. For instance, development organisations and research institutes in the consortium, unlike university-based academics, did not have institutional incentives to publish academically. Partly this has to do with the length of time it takes to go through long peer-review and publication processes as well as trying to navigate unknown journal intranet systems. In the case of ECID, this led to global North colleagues, who had more institutional support and familiarity with publishing processes, leading and co-authoring publications. This did not happen in self-published research reports, which were autonomously written by each country team.

Finally, a conflict of interest also arose in how the results were interpreted and used, which is a constant trade-off in multi-country research. The solution is to find a "hybrid

identity" of results which are relevant for the country of study but also internationally and in a comparative perspective. Getting this balanced approach right in a report is a precious academic and writing skill.

### 3.8. Making Research Outputs Available for Free, and in Different Languages

Many researchers worldwide, especially those in poorer countries, poorer universities, and those who are researching outside academia, simply cannot afford the cost of accessing journals that charge a fee to access research papers. This serves to reproduce knowledge hegemonies that favour the global North.

On a positive note, sharing research outputs freely is gradually becoming a common practice worldwide. This practice is especially relevant for decolonisation processes and forms the final component that should be considered when seeking to shift power in research.

However, one of the challenges with open access publications is that they place the burden of financial responsibility on the writer. These costs are usually quite high, making open access publishing only accessible to more wealthy academics and academic institutes. This, in turn, can be read as an elitist and colonial practice that also perpetuates knowledge hegemonies. This requires finding alternative approaches to publishing and accessing journal articles.

One alternative approach is to upload the last draft of the accepted manuscript in research repositories such as *Academia* and *ResearchGate* where they are freely available. In ECID, the literature reviews and the larger research reports or working papers that were produced to inform more academic publications were made available on the ECID webpage.

Finally, another challenge relates to the issue of language diversity, which is particularly challenging. On the one hand, using English as a bridging language means that the publication can have a greater reach. On the other hand, English keeps the colonial legacy in place by replacing other languages [25]. To address this challenge, some options to consider include publishing in English but with journals that welcome publications in other languages and/or that translate abstracts. Another option is to publish different research aspects in different languages.

### 4. Conclusions

The current research ecosystem does not lend itself to shifting power in research and building more equitable research partnerships. While, comparatively, ECID was a mid-sized research project (its initial budget was of around GBP 350,000), this case study demonstrated that it is possible to integrate decolonial research approaches into practice at scale. As such, it offers a model towards decolonising research and research ethics in international research. It did so by addressing what we consider the three hardest structural power issues—a shared research design, a shared budget, and shared publications—and by suggesting eight practical steps towards a model for shifting power in research and decolonising research ethics:

1. Build a truly global partnership structure;
2. Decentralise budgets towards the global South;
3. Build a multi-country and hybrid research team;
4. Build a multi-country and hybrid ethics panel;
5. Co-design research;
6. Cite global South authors and perspectives, and oral sources;
7. Publish in global South journals;
8. Make research outputs available for free, and in different languages.

It is important to recognise that although this approach was developed through the ECID research, it also came with several challenges. For example, decentralising research budgets risks not being able to deliver on financial commitments to partner organisations. In this study, the ECID programme closed two years early due to a decision by the UK Government to cut foreign aid funding. This directly impacted partners and their ability

to produce research outputs. As a result, steps 7 and 8 that have been proposed in this model have been delayed and, as of yet, have not been fully carried out. The decision also had financial consequences on the commitments made to partner organisations, which also came with ethical fractures.

Another example relates to the remit of the ethics panel which, in part, was to ensure the quality of the research. However, introducing diverse research teams also introduced methodological diversity into the global team in which some country research teams took a more quantitative approach to their research while others took a more qualitative approach. This introduced a different type of power dynamic relating to politics and hierarchies in evidence which had to be addressed. In addition, there were different understandings of what constituted 'quality evidence' which also had to be explored.

Challenges also emerged around the process that was used to co-design the research which focused more on learning around the research process. In keeping with a more decolonial approach to ethics, it should have included more learning around the cultural and contextual aspects of the research. This is especially important if researchers do not come from the country in which the research is being carried out. In ECID, this was less relevant as all the researchers were from the participating countries.

Similarly, there were challenges in relation to the power dynamics within the research, which a decolonial approach also requires one to consider. As such, more attention should be paid to other inequalities beyond coloniality. In research, gender and age imbalances are often prominent, which maters in certain social contexts. This requires more consideration than was given within ECID.

From an ethical perspective, we suggest that "ethics" become a daily struggle to address power relations and that it become synonymous with "decolonising" approaches in research. Researchers should be aware of the legal-practical debate around ethics, but also of the project-structural debate, as these aim to move ethics beyond the single research project towards addressing systemic power imbalances such as decoloniality.

Ethical frameworks and codes are important to advance a wider vision of ethics, but they but will not be enough to achieve knowledge justice if they do not challenge the research structures and wider research ecosystems that support knowledge hegemonies. A single research project can be perfectly ethical in terms of how informed consent is administered, maintaining confidentiality and a person's freedom to participate in the research, yet it can still reinforce the unequal research structures that prevent building equitable North–South research partnerships. Doing research ethically also requires addressing global inequalities.

To achieve knowledge justice requires that development donors fund more decentralised research and research ideas coming from universities and non-academic research institutes based in the global South. It also requires ethical review bodies becoming more geographically and professionally diverse and academics from the global North and South being able to publish and access journals more equitably. Unless the systems that nurture knowledge production become more equal, no North–South research project alone—whichever code, framework, or review board it follows—will be ethical.

Consequently, it is imperative that we apply a decolonial lens to our research codes, frameworks, review boards, and across the research cycle. We believe that the 8-step model proposed in this paper can be adopted, fully or in part, in the research policies held by both professional and academic entities. Even if universities are not involved in North–South research, they should start including practitioners in their ethical and research processes to ensure more contextually appropriate and rooted research. Likewise, organisations without ethics review boards and processes can follow this model to ensure that they are carrying out research ethically.

Finally, there is also a need for further research to track progress in relation to the state of global research ecosystems. Fransman [26] offers a model towards which this can be achieved. As we work towards knowledge justice, we should be reminded of remarks made by Dr Toby Oshodi (from Lagos State University) in a knowledge exchange seminar

hosted by Christian Aid in 2021, in which he said: "decolonising is not disconnecting [from European partners]". It is about making global funds and research systems work towards equality, not against it.

**Author Contributions:** Conceptualization, methodology, software, validation, formal analysis, investigation, resources, data curation, visualization, supervision, project administration, funding acquisition, M.J.C.S., T.A. and C.B.; writing—original draft preparation, M.J.C.S.; writing—review and editing, T.A. and C.B. All authors have read and agreed to the published version of the manuscript.

**Funding:** This research was funded by the UK Government, under its programme UK Aid Connect.

**Informed Consent Statement:** Not applicable.

**Data Availability Statement:** For research outputs and context, please visit: https://evidenceforinclusion.org/research/ (accessed on 20 March 2022). There are no publicly archived datasets.

**Acknowledgments:** We would like to thank the ECID team and the research partners in the four countries. For a list of participants, please visit: https://evidenceforinclusion.org/research/ (accessed on 20 March 2022).

**Conflicts of Interest:** The authors declare no conflict of interest. The funders had no role in the design of the study; in the collection, analyses, or interpretation of data; in the writing of the manuscript; or in the decision to publish the results.

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
