# Peer review of "Towards Decolonising Research Ethics: From One-off Review Boards to Decentralised North–South Partnerships in an International Development Programme"

_education, doi:10.3390/educsci12040236_

Round 1

Reviewer 1 Report

  1. Moderate English changes required
  2. Reviewer suggests to write again the abstract within the limit of 200 words including more literature background.
  3. The research design, questions, hypotheses and methods must be written again.
  4. Methodology and Overview of the Case Study (ECID) section must be expanded.
  5. empirical research results must be more clearly presented.

Author Response

Please, see attached responses to the review.

With thanks,

The authors

Reviewer 2 Report

The article “Toward decolonizing research ethics,” based on a case study, proposes a turning point on both technical as well political reading of ethics in science. Although built from a social sciences approach, several points can be extrapolated to a more complex discussion in how inequalities north-south are reproduced under a narrow understanding of what research ethics is for. In short, the paper refreshes this discussion, argues for an ethical view committed to diversity, local culture, opens to co-design creation - far away from an instrumentalized procedural ethics-, and proposes 8-shifting points. While 3 of them were not extensively put in practice, and despite limitations indicated by authors, such as a single case based on the author’s self-ethnography, the work is powerfully relevant to epistemology and ethics of science.     

As proposed by the authors, all the findings and discussions guide their understanding of what decolonizing ethics in research means to them. Hence, the philosophical depth and its sophistication came from their empirical evidence. The inductivity does not compromise the conclusions. In fact, it can be perceived as a strong point for future debates on the field. The authors clearly show their point of view and give their experience as a case study to north-south multi-countries research, especially when adopting a decolonial approach.

The methodology is appropriate to the objectives and research question. The article is well-structured, clear, and well-wrote. There is a critical and solid argument against hegemonic practices in north-south research projects. Also, an interesting proposal on how the ethical parameters and procedures for a non-asymmetric power multi-country research should be, political and ethical committed with partners’ local realities.  

For all these arguments, I recommend to editors publish this manuscript in its terms.

Author Response

(The authors gave the same response as above.)

Reviewer 3 Report

Thanks for the opportunity to read this paper, I found it to be well written, interesting, and straightforward.

Maybe I'm just not the target audience here, but I'd really like to see some of the terms/phrases used in this paper defined and explained. Even those I am familiar with, it would help the audience to understand what the authors mean when they use these terms. 

Terms that I'd like to see defined: "global North" "global South" "decolonize" 

Overall I really like the ideas presented in this paper and I appreciate the case study as an example.  One thing that I think could be discussed in greater detail here are the power dynamics of the researchers themselves in relation to their research subjects and how to address those potential biases. Similarly, the power dynamics between the PIs and any graduate students/hired staff.

Not specifically addressed is that much of what is covered here (IMO) are not just colonial power structures, but also conflicts of interest. I wonder if adding some discussion about using these approaches to reduce conflicts of interest would add greater applicability to this paper.  

Finally, I wonder if its worth addressing how to codify or enforce some of the approaches here, as they would improve overall research in terms both reducing conflicts of interest, as well as decolonizing science and scholarship.

Author Response

(The authors gave the same response as above.)
